# 🌍WONDERZOOM:
# MULTI-SCALE 3D WORLD GENERATION

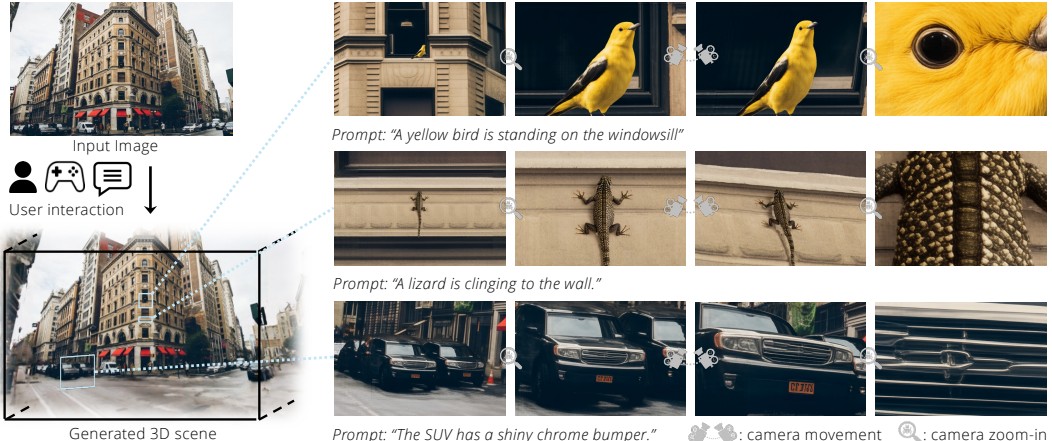

Figure 1: **Multi-scale 3D world generation from a single image.** WonderZoom enables interactive exploration across spatial scales. Users can zoom into any region and specify prompts to generate new fine-scale content while maintaining cross-scale consistency. Here we show three zoom-in sequences.

## ABSTRACT

We present WonderZoom, a novel approach to generating 3D scenes with contents across multiple spatial scales from a single image. Existing 3D world generation models remain limited to single-scale synthesis and cannot produce coherent scene contents at varying granularities. The fundamental challenge is the lack of a scale-aware 3D representation capable of generating and rendering content with largely different spatial sizes. WonderZoom addresses this through two key innovations: (1) scale-adaptive Gaussian surfels for generating and real-time rendering of multi-scale 3D scenes, and (2) a progressive detail synthesizer that iteratively generates finer-scale 3D contents. Our approach enables users to "zoom into" a 3D region and auto-regressively synthesize previously non-existent fine details from landscapes to microscopic features. Experiments demonstrate that WonderZoom significantly outperforms state-of-the-art video and 3D models in both quality and alignment, enabling multi-scale 3D world creation from a single image. We show video results and an interactive viewer of generated multi-scale 3D worlds in `https://wonderzoom.github.io/`.

## 1 INTRODUCTION

3D world generation has emerged as a transformative capability in computer vision, enabling the synthesis of immersive environments from minimal input (Yu et al., 2024; 2025; Chung et al., 2023; Höllein et al., 2023; Fridman et al., 2023; Liu et al., 2021). However, despite the inherently multi-scale nature of real-world scenes, existing approaches remain fundamentally constrained to single-scale generation. They can produce landscape-level environments and room-scale scenes, but fail to synthesize coherent content across multiple spatial scales, e.g., a tiny ladybug lying on a sunflower in a vast field. This limitation prevents existing approaches from generating rich, detailed worlds that span from panoramic vistas down to microscopic surface details, restricting their applicability for interactive exploration and content creation.

The fundamental challenge underlying this limitation is the absence of a scale-adaptive 3D representation suitable for scene generation. Traditional Level-of-Detail (LoD) representations (Luebke et al., 2002) were designed for efficiently rendering pre-existing graphics content, where all geometric details are known in advance. Recent hierarchical representations like Hierarchical 3D Gaussian Splatting (Kerbl et al., 2024) and Mip-NeRF (Barron et al., 2021) extend these principles to neural reconstruction, efficiently encoding scenes at multiple scales. But critically, they still assume access to complete multi-scale image data upfront for one-pass optimization. Both paradigms, rendering and reconstruction, fundamentally conflict with *generation*, where images do not exist a priori and must be synthesized progressively. In generation, we must create coarse-scale content first, then iteratively synthesize finer details conditioned on both the coarser structure and user-specified prompt and regions of interest. This requires representations that can grow dynamically as new fine-scale content is generated, not static hierarchies optimized with complete supervision. Current generation methods (Yu et al., 2024; 2025) sidestep this challenge entirely by restricting themselves to single scales, while naive application of existing hierarchical representations would demand generating all scales simultaneously, which is a computationally intractable approach that violates the inherent coarse-to-fine nature of multi-scale synthesis.

To address this challenge, we propose WonderZoom, a novel framework for multi-scale 3D world generation from a single image. Our approach introduces two key technical innovations: (1) *scale-adaptive Gaussian surfels*, a dynamically updatable hierarchical representation that, unlike existing multi-scale methods, supports incremental refinement as new content is generated. It allows adding arbitrary levels of detail without re-optimization, and (2) a *progressive detail synthesizer* that iteratively generates fine-grained 3D structures conditioned on both coarser scales and user-specified regions and viewpoints. These components work synergistically: the scale-adaptive representation provides a persistent 3D canvas that grows in detail over time, while the synthesizer produces coherent multi-scale content through a controlled coarse-to-fine generation process. By enabling dynamic updates to the 3D representation as new scales are synthesized, WonderZoom fundamentally shifts from the reconstruction paradigm to multi-scale generation, overcoming the computational and architectural barriers that constrain existing methods to single scales.

Our approach enables users to interactively "zoom into" any region of the generated 3D scene, triggering autoregressive synthesis of previously non-existent details, e.g., from an entire landscape down to microscopic surface features. Unlike traditional multi-resolution rendering that simply reveals pre-existing details, WonderZoom *generates* new content on-demand, creating coherent structures that were never part of the original input or coarse generation. This capability allows infinite exploration of generated worlds at arbitrary levels of detail. In summary, our contributions are threefold:

- We propose WonderZoom, the first approach to enable multi-scale 3D world generation from a single image, supporting seamless transitions from macro to micro scales.

- We introduce scale-adaptive Gaussian surfels, a dynamically updatable representation that grows incrementally with newly generated finer-scale content, while maintaining real-time rendering performance.

- We demonstrate and evaluate multi-scale 3D generation across diverse scenarios including natural environments, villages, and urban scenes, achieving consistent quality across scale transitions while significantly outperforming state-of-the-art video and 3D generation models in both perceptual quality and prompt alignment.

## 2 RELATED WORK

**3D World Generation.** Early 3D scene generation methods focused on novel view synthesis from a single image, constructing renderable representations like layered depth images (Tulsiani et al., 2018; Shih et al., 2020), radiance fields (Yu et al., 2020; Trevithick & Yang, 2020; Szymanowicz et al., 2024), multi-plane images (Tucker & Snavely, 2020; Zhou et al., 2018), and point features (Niklaus et al., 2019; Wiles et al., 2020), though these only supported small viewpoint changes from the input. Later works explored generating more significant viewpoint changes and multiple connected scenes. Infinite Nature (Liu et al., 2021) and its follow-ups (Li et al., 2022; Chai et al., 2023; Cai et al., 2023) pioneered perpetual view generation for natural scenes with a neural renderer. Recent methods (Liang

et al., 2025; Yang et al., 2025; Team et al., 2025; Zhou et al., 2024; Li et al., 2024) expanded this capability to explicit 3D, e.g., SceneScape (Fridman et al., 2023) and Text2Room (Höllein et al., 2023) generate meshes from text prompts, WonderJourney (Yu et al., 2024) and WonderWorld (Yu et al., 2025) creates diverse connected 3D scenes using LLMs and point-based representations, LucidDreamer (Chung et al., 2023) and CAT3D (Gao et al., 2024) focus on room-scale environments with 3D Gaussian splatting. Another line of work specializes in city-scale generation (Lin et al., 2023; Xie et al., 2024a;b; Engstler et al., 2025), producing large-scale 3D Gaussian splatting representations of urban environments. However, these methods operate at a single spatial scale aligned with their input——generating either landscapes, rooms, or cities, but not content across scales. In contrast, we enable multi-scale 3D generation where users can progressively zoom into any region to synthesize entirely new content at finer scales, creating details that were never visible or implied in the original input image.

**Multi-scale 3D Representations.** Classical computer graphics has long addressed multi-scale rendering through Level-of-Detail (LoD) techniques (Luebke et al., 2002), which adaptively select geometric complexity based on viewing distance, and mipmapping, which precomputes texture pyramids for efficient anti-aliased rendering. These traditional methods assume all geometric and texture details exist upfront, making them suitable only for rendering pre-authored content, not for progressive generation. Recent neural 3D reconstruction methods have incorporated similar multi-scale principles, e.g., Mip-NeRF (Barron et al., 2021) introduces integrated positional encoding to handle scale ambiguity, with extensions like Mip-NeRF 360 (Barron et al., 2022) and Zip-NeRF (Barron et al., 2023) improving unbounded scene representation. In the Gaussian splatting (Kerbl et al., 2023) domain, Mip-Splatting (Yu et al., 2023) addresses aliasing through 3D smoothing filters, while Hierarchical 3D Gaussian Splatting (Kerbl et al., 2024) builds explicit LoD hierarchies for efficient rendering. Octree-GS (Ren et al., 2024a) and Scaffold-GS (Lu et al., 2024) use spatial hierarchies to manage primitives across scales. However, both traditional LoD and these neural hierarchical representations share a critical limitation: they are fundamentally designed for scenarios where content at all scales is known: either pre-authored (traditional LoD) or reconstructed from complete multi-scale image supervision (neural methods). This paradigm is incompatible with generation, where fine-scale content must be synthesized progressively without pre-existing data. Our approach addresses this gap by organically integrating a scale-adaptive representation that can be dynamically refined with a progressive generation pipeline.

**Controllable Content Synthesis.** Controllable video generation methods have made significant strides in conditional synthesis, accepting camera trajectories (He et al., 2024; Ren et al., 2025), depth maps (Zhang et al., 2023), or semantic masks as inputs to guide generation. However, these approaches cannot perform multi-scale generation due to the absence of training data that captures coherent content across vastly different spatial scales. Super-resolution techniques have evolved from 2D image enhancement to 3D domains, including mesh refinement, point cloud upsampling (Zhang et al., 2022), and neural field super-resolution (Wang et al., 2022). Yet these methods focus on sharpening and refining pre-existing content rather than generating entirely new cross-scale structures. A recent work, Generative Powers of Ten (Wang et al., 2024b), demonstrates infinite zoom generation by jointly sampling multiple scales through coordinated diffusion processes, though this remains limited to 2D images. Hierarchical generation approaches like Progressive GANs (Karras et al., 2021) and cascaded diffusion models (Ho et al., 2022) synthesize content at increasing resolutions through staged refinement. Our approach uniquely extends these capabilities to 3D, combining controllable generation with true multi-scale synthesis—enabling users to interactively zoom into any region and generate coherent new content across vastly different spatial scales, from environmental to microscopic levels that never existed in the original input.

## 3    APPROACH

**Formulation.** We target *multi-scale 3D world generation* from a single image. Given an input image $\mathbf{I}_0$ and a sequence of user-specified prompts $\{\mathcal{U}_1, \ldots, \mathcal{U}_n\}$ with corresponding camera viewpoints $\{\mathbf{C}_0, \ldots, \mathbf{C}_n\}, \mathbf{C}_i \in \mathbb{R}^{4 \times 4}$ that progressively zoom into regions of interest, our goal is to generate a sequence of 3D scenes $\{\mathcal{E}_0, \mathcal{E}_1, \ldots, \mathcal{E}_n\}$ at increasing spatial granularities. Here, $\mathcal{E}_0$ represents the initial 3D scene reconstructed from the input image $\mathbf{I}_0$, while each subsequent scene $\mathcal{E}_i$ $(i > 0)$ represents finer-scale content that is spatially contained within the previous scene $\mathcal{E}_{i-1}$, creating a

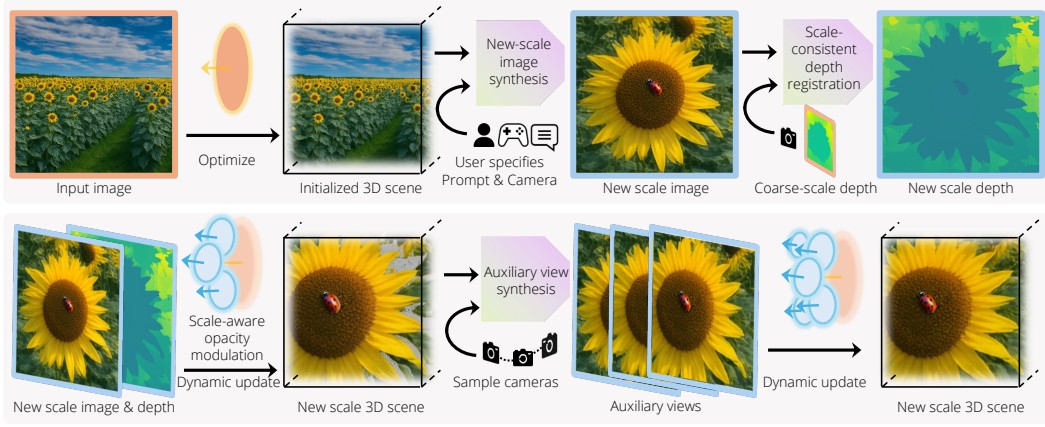

Figure 2: **WonderZoom overview.** From an input image, we first reconstruct an initial 3D scene. Users specify prompts and camera viewpoints to generate finer-scale content. Our progressive detail synthesizer creates new-scale images, registers depth to maintain geometric consistency, and synthesizes auxiliary views for complete 3D scene creation. Our scale-adaptive Gaussian surfels enable dynamic updates without re-optimization, seamlessly integrating new content while preserving real-time rendering.

nested hierarchy where zooming reveals newly synthesized details rather than pre-existing geometry. This process can be repeated multiple times from the same initial image $\mathbf{I}_0$ with different camera sequences and prompt sequences. Figure 1 illustrates this capability, where we demonstrate three distinct zoom sequences from a single input.

**Challenges.** The major technical bottleneck preventing multi-scale generation is the lack of scale-adaptive 3D representations suitable for generation. Existing multi-scale representations, from classical Level-of-Detail techniques to recent hierarchical methods like Hierarchical 3D Gaussian Splatting (Kerbl et al., 2024), are designed for either rendering pre-existing graphics content or reconstruction with complete multi-scale image supervision available upfront. However, generation imposes fundamentally different requirements: we need to create coarse-scale content $\mathcal{E}_{i-1}$ first, then iteratively synthesize finer details $\mathcal{E}_i$ conditioned on both the coarser structure $\mathcal{E}_{i-1}$ and user-specified prompts $\mathcal{U}_i$ and regions of interest defined by $\mathbf{C}_i$. This demands representations that can grow dynamically as new scales are generated while maintaining real-time rendering capability, a capability absent in existing methods that assume static, pre-optimized hierarchies. Another challenge lies in synthesizing semantically meaningful content that follows user prompts $\mathcal{U}_i$ while maintaining geometric and appearance consistency with previous scales $\mathcal{E}_{i-1}$. Unlike simple super-resolution that merely enhances existing details, we may need to generate entirely new structures (e.g., a bird or a lizard as in Figure 1) that were not implied in the coarser representation.

**Overview.** We propose WonderZoom to enable multi-scale 3D world generation through two key technical innovations. To address the first challenge, we introduce *scale-adaptive Gaussian surfels* (Sec. 3.1) that allow dynamic updates without re-optimization. This representation enables adding arbitrarily many scales $\mathcal{E}_i$ while maintaining real-time rendering capability at any scale, as new finer-scale surfels can be seamlessly integrated into the existing hierarchy without modifying coarser levels. To address the second challenge, we design a *progressive detail synthesizer* (Sec. 3.2) that generates new fine-grained 3D structures $\mathcal{E}_i$ from user prompts $\mathcal{U}_i$ while ensuring consistency with the previous scale $\mathcal{E}_{i-1}$. The synthesizer leverages the coarse geometry as spatial conditioning to guide the generation of coherent fine-scale content, going beyond simple super-resolution to create semantically meaningful details. We show an illustration of our framework in Figure 2. We summarize the complete multi-scale generation control loop in Algorithm 1 in supplementary material.

### 3.1 SCALE-ADAPTIVE GAUSSIAN SURFELS

**Definition.** We introduce scale-adaptive Gaussian surfels to represent our multi-scale scenes $\{\mathcal{E}_0, \ldots, \mathcal{E}_n\}$. Formally, we model the scenes as a radiance field represented by a set of Gaussian surfels $\{g_j\}_{j=1}^N$. Each surfel is parameterized as $g = \{\mathbf{p}, \mathbf{q}, \mathbf{s}, o, \mathbf{c}, s^{\text{native}}\}$, where $\mathbf{p}$ denotes

the 3D spatial position, $\mathbf{q}$ denotes the orientation quaternion, $\mathbf{s} = [s_x, s_y]$ denotes the scales of the $x$-axis and $y$-axis, $o$ denotes the opacity, and $\mathbf{c}$ denotes the view-independent RGB color. The Gaussian kernel follows the same formulation as in prior work (Yu et al., 2025), with covariance matrix $\mathbf{\Sigma} = \mathbf{Q}\mathrm{diag}(s_x^2, s_y^2, \epsilon^2)\mathbf{Q}^T$ where $\mathbf{Q}$ is the rotation matrix obtained from $\mathbf{q}$ and $\epsilon$ is a small thickness parameter. The key addition is $s^{\mathrm{native}}$, the native scale at which the surfel was created, which enables scale-aware rendering as we describe later. In WonderZoom, we sequentially generate each scene, starting from $\mathcal{E}_0$ and progressively adding finer-scale content through $\mathcal{E}_n$. This demands our representation to satisfy two requirements: (1) capable of dynamic updates given new scale images $\mathbf{I}_i$ at viewpoints $\mathbf{C}_i$ without re-optimizing existing surfels, and (2) supporting real-time rendering at any observation scale.

**Dynamic updating.** The core idea of our dynamic representation is that we incrementally add new surfels to represent each new scale without modifying existing ones. When we create the initial scene $\mathcal{E}_0$ from the input image $\mathbf{I}_0$, we generate $N_0$ surfels to represent the coarse-scale geometry and appearance. When we subsequently generate the finer-scale scene $\mathcal{E}_1$ from a zoomed-in view $\mathbf{I}_1$ at camera $\mathbf{C}_1$, we add $N_1$ new surfels to the representation, resulting in a total of $N = N_0 + N_1$ surfels. This process continues: when generating $\mathcal{E}_i$, we add $N_i$ new surfels, bringing the total to $N = \sum_{k=0}^{i} N_k$. Crucially, the surfels from previous scales remain unchanged: we only append new surfels that capture the finer details visible at the current scale. This additive mechanism naturally enables dynamic updates: each new scale simply extends the existing representation rather than requiring global re-optimization, allowing the multi-scale world to grow organically as users explore different regions at increasing levels of detail.

**Scale-aware opacity modulation for real-time rendering of multi-scale scenes.** Since we represent multi-scale content with surfels across different scales, the same surface may be covered by multiple layers of surfels from $\mathcal{E}_0$ through $\mathcal{E}_i$. Directly rendering all surfels would cause aliasing and reduce rendering speed. To address this, we introduce scale-aware opacity modulation based on each surfel's native scale:

$$s^{\mathrm{native}} = \frac{d^{\mathrm{native}}}{\sqrt{f_x^{\mathrm{native}} f_y^{\mathrm{native}}}} \tag{1}$$

where $d^{\mathrm{native}}$ is the surfel's depth relative to $\mathbf{C}_i$ (the camera view where the surfel was created) and $f_x^{\mathrm{native}}, f_y^{\mathrm{native}}$ are the focal lengths of $\mathbf{C}_i$. During rendering at camera $\mathbf{C}^{\mathrm{render}}$, we compute the current rendering scale $s^{\mathrm{render}} = d^{\mathrm{render}}/\sqrt{f_x^{\mathrm{render}} f_y^{\mathrm{render}}}$ for each surfel. For surfels at intermediate scales $(0 < i < n)$, we define parent and child scale bounds: $s^{\mathrm{parent}} = d^{\mathrm{parent}}/\sqrt{f_x^{\mathrm{parent}} f_y^{\mathrm{parent}}}$ where $d^{\mathrm{parent}}$ and $f^{\mathrm{parent}}$ are defined relative to $\mathbf{C}_{i-1}$, and $s^{\mathrm{child}} = d^{\mathrm{child}}/\sqrt{f_x^{\mathrm{child}} f_y^{\mathrm{child}}}$ where $d^{\mathrm{child}}$ and $f^{\mathrm{child}}$ are defined relative to $\mathbf{C}_{i+1}$. The rendered opacity is then modulated as:

$$\tilde{o} = o \cdot \alpha, \quad \text{where } \alpha = \begin{cases} 1 & \text{if no parent exists and } s^{\mathrm{render}} \geq s^{\mathrm{native}} \\ \frac{\log(s^{\mathrm{parent}}) - \log(s^{\mathrm{render}})}{\log(s^{\mathrm{parent}}) - \log(s^{\mathrm{native}})} & \text{if } s^{\mathrm{parent}} \geq s^{\mathrm{render}} \geq s^{\mathrm{native}} \\ \frac{\log(s^{\mathrm{render}}) - \log(s^{\mathrm{child}})}{\log(s^{\mathrm{native}}) - \log(s^{\mathrm{child}})} & \text{if } s^{\mathrm{native}} \geq s^{\mathrm{render}} \geq s^{\mathrm{child}} \\ 1 & \text{if no child exists and } s^{\mathrm{render}} \leq s^{\mathrm{native}} \\ 0 & \text{otherwise.} \end{cases} \tag{2}$$

This design ensures surfels are most visible at their native scale ($\alpha = 1$ when $s^{\mathrm{render}} = s^{\mathrm{native}}$) and fade smoothly when viewed at different scales. Notably, surfels at the coarsest scale ($i = 0$) remain fully visible when zoomed out, while surfels at the finest scale ($i = n$) remain fully visible when zoomed in, ensuring complete scene coverage at all observation scales.

**Proposition 1 (Seamless Scale Transition).** Our scale-aware opacity modulation ensures smooth visual transitions between adjacent scales without discontinuities. Specifically, consider two surfels $g_j$ and $g_k$ located at the same 3D position $\mathbf{p}$ but created at adjacent scales $\mathcal{E}_{i-1}$ and $\mathcal{E}_i$ respectively. When the rendering scale $s^{\mathrm{render}}$ transitions between their native scales, i.e., when $s^{\mathrm{render}} \in [s_k^{\mathrm{native}}, s_j^{\mathrm{native}}]$, the sum of their modulated opacity weights satisfies:

$$\alpha_k(s^{\mathrm{render}}) + \alpha_j(s^{\mathrm{render}}) = 1. \tag{3}$$

This property holds because the linear interpolation in log space for $g_k$ decreasing from its native scale matches exactly with the complementary interpolation for $g_j$ increasing toward its child scale bound. As a result, the total contribution from overlapping surfels at different scales remains constant during zoom operations, eliminating popping artifacts and ensuring visually continuous scale transitions. This partition of unity is fundamental to maintaining coherent appearance as users navigate across the multi-scale hierarchy.

**Optimization.** Our scale-aware opacity modulation preserves the differentiability of the rendering pipeline, thereby we use gradient-based optimization for surfel parameters. When creating surfels for a new scale $\mathcal{E}_i$ from image $\mathbf{I}_i$, we generate pixel-aligned surfels following the same approach as prior work (Yu et al., 2025), where each surfel corresponds to a pixel in $\mathbf{I}_i$. We also follow the same geometry-based initialization: each surfel's position $\mathbf{p}$ is initialized using the estimated depth map via back-projection, orientation $\mathbf{q}$ from the estimated surface normal, and scales $\mathbf{s}$ according to the Nyquist sampling theorem to ensure appropriate coverage without excessive overlap. The color $\mathbf{c}$ is initialized from the corresponding pixel RGB values, the native scale $s^{\text{native}}$ is computed based on the creation viewpoint $\mathbf{C}_i$, and opacity is initialized to $o = 0.1$ for stable optimization. We then optimize the opacity, orientation, and scales (while keeping positions, colors, and native scales fixed) using Adam (Kingma & Ba, 2014) with a photometric loss $\mathcal{L} = 0.8L_1 + 0.2L_{\text{D-SSIM}}$ (Kerbl et al., 2023) against the input image $\mathbf{I}_i$. This lightweight optimization refines the surfel geometry while preserving the multi-scale structure.

### 3.2 Progressive Detail Synthesizer

**Goal.** Given the coarse-scale scene $\mathcal{E}_{i-1}$, a target camera viewpoint $\mathbf{C}_i$, and a user prompt $\mathcal{U}_i$, our goal is to generate an image $\mathbf{I}_i$ and its corresponding depth map $\mathbf{D}_i$ that are geometrically consistent with $\mathcal{E}_{i-1}$ while incorporating the content specified in $\mathcal{U}_i$. Note that $\mathcal{U}_i$ may describe entirely new structures not visible or implied in $\mathcal{E}_{i-1}$ (e.g., a ladybug on a sunflower), requiring our approach to go beyond simple super-resolution to synthesize semantically meaningful content. Since we aim to generate a complete 3D scene $\mathcal{E}_i$ that can be rendered from varying viewpoints, we additionally generate a set of auxiliary images $\{\mathbf{I}_i^k\}_{k=1}^K$ from neighboring viewpoints to augment $\mathbf{I}_i$, enabling optimization of a more complete 3D structure that extends beyond the single input view. This subsection describes our three-stage pipeline: new scale image generation from the coarse scene and prompt, scale-consistent depth registration to maintain geometric coherence, and auxiliary view synthesis for complete 3D reconstruction.

**New scale image synthesis.** To generate the finer-scale image $\mathbf{I}_i$, we begin by rendering a coarse observation from the previous scale: $\mathbf{O}_i = \text{render}(\mathcal{E}_{i-1}, \mathbf{C}_i)$, where $\mathbf{C}_i$ has a larger focal length than $\mathbf{C}_{i-1}$ to zoom into the region of interest. Since $\mathbf{O}_i$ is obtained through direct zoom-in rendering and thus lacks fine details, we apply extreme super-resolution to synthesize high-frequency content. However, extreme zoom ratios require additional semantic guidance beyond what is visible in $\mathbf{O}_i$. We therefore extract semantic context from the previous scale using a vision-language model (VLM): $\mathcal{S} = \text{VLM}(\mathbf{O}_{i-1})$, where $\mathbf{O}_{i-1}$ is the rendered image at the previous scale. The super-resolved image is then generated as $\mathbf{I}_i' = \text{SR}(\mathbf{O}_i, \mathcal{S})$, conditioned on both the coarse observation and semantic context. To incorporate user-specified content $\mathcal{U}_i$ that may include entirely new structures absent in $\mathcal{E}_{i-1}$, we apply a controllable image editing model: $\mathbf{I}_i = \text{Edit}(\mathbf{I}_i', \mathcal{U}_i)$. This two-stage approach—super-resolution followed by semantic editing—enables both faithful detail enhancement of existing structures and insertion of novel content specified by the user.

**Scale-consistent depth registration.** To estimate a depth map $\mathbf{D}_i$ that maintains geometric consistency with $\mathcal{E}_{i-1}$, we employ a multi-stage registration approach. First, we render a target depth map from the existing geometry: $\mathbf{D}_i^{\text{target}} = \text{render\_depth}(\mathcal{E}_{i-1}, \mathbf{C}_i)$, which provides sparse depth values for regions visible in the previous scale. We then fine-tune a monocular depth estimator $\mathcal{D}_\theta$ to align its predictions with this target depth by minimizing:

$$\mathcal{L}_{\text{depth}} = \frac{\sum_{u,v} \|\mathbf{D}_i^{\text{target}}(u,v) - \mathcal{D}_\theta(\mathbf{I}_i)(u,v)\| \cdot m(u,v)}{\sum_{u,v} m(u,v)}, \tag{4}$$

where $m(u,v) = 1$ if $\mathbf{D}_i^{\text{target}}(u,v)$ is defined, and $m(u,v) = 0$ for undefined regions due to zoom-in effect. This fine-tuning ensures that the estimated depth $\mathbf{D}_i = \mathcal{D}_\theta(\mathbf{I}_i)$ aligns with the coarse geometry

while still predicting reasonable depths for newly visible regions. To further refine the registration, we apply segment-wise depth alignment using SAM-generated masks to correct for local depth inconsistencies as in prior work (Yu et al., 2024; 2025), and for any newly added structures from the editing stage (e.g., the ladybug in Figure 2), we use Grounded SAM (Ren et al., 2024b) to isolate these regions and estimate their depth while maintaining consistency with surrounding geometry.

**Auxiliary view synthesis.** While $\mathbf{I}_i$ provides detailed content at the target viewpoint $\mathbf{C}_i$, a single image is insufficient to reconstruct a complete 3D scene that can be rendered from arbitrary viewpoints. To address this, we synthesize auxiliary views $\{\mathbf{I}_i^k\}_{k=1}^K$ from neighboring camera positions using a camera-controlled video diffusion model. We first render conditioning frames from the current partial scene: $\{\mathbf{O}_i^k\} = \{\text{render}(\mathcal{E}_i^{\text{partial}}, \mathbf{C}_i^k)\}_{k=1}^K$, where $\mathcal{E}_i^{\text{partial}}$ is the initial scene constructed from $\mathbf{I}_i$ alone, and $\{\mathbf{C}_i^k\}$ are camera viewpoints sampled around $\mathbf{C}_i$. Along with these frames, we generate corresponding masks $\{\mathbf{M}_i^k\}$ indicating regions requiring synthesis (e.g., occluded areas not visible in $\mathbf{I}_i$). The video diffusion model then generates temporally consistent frames: $\{\mathbf{I}_i^k\} = \text{VideoDiff}(\{\mathbf{O}_i^k\}, \{\mathbf{M}_i^k\})$, conditioned on the partial observations and masks. We then leverage a video depth model to estimated depth $\{\mathbf{D}_i^k\}$ for these generated frames, and the resulting image-depth pairs are used to optimize a more complete 3D scene following the same optimization procedural as described in Sec. 3.1. This auxiliary view synthesis enables us to construct complete 3D scenes $\mathcal{E}_i$ that extend beyond the single input view while maintaining coherence with the generated content. In practice, we also apply it to help generate the coarsest-scale scene $\mathcal{E}_0$.

## 4 EXPERIMENTS

In our experiments, we evaluate WonderZoom on multi-scale world generation and compare it to existing methods. We also perform ablation studies to analyze WonderZoom.

**Baselines.** We are not aware of any prior method that allows multi-scale 3D scene generation. Therefore, we consider state-of-the-art methods in general-purpose 3D scene generation including WonderWorld (Yu et al., 2025) and HunyuanWorld (Team et al., 2025). Besides 3D-based approaches, we further include state-of-the-art camera-controlled video generation models, including Gen3C (Ren et al., 2025) and Voyager (Huang et al., 2025). We use these baselines' official codes for comparison.

**Test examples.** For comparison with the baselines, we collect publicly available real images and generate synthetic images as our testing examples, and we also use examples from Wang et al. (2024b). We use 6 test examples spanning diverse scene types such as a field, a city, a forest, and underwater. Among them, a sunflower image and a coral image are synthetic, and all others are real images. For each test example, we generate 4 scenes besides the input scene, i.e., we generate $\{\mathcal{E}_0, \cdots, \mathcal{E}_4\}$. For a fair comparison, we use fixed camera paths and the same text prompts for all methods.

**Metrics.** For quantitative comparison, we adopt the following evaluation metrics: (1) To evaluate the alignment of generated scenes w.r.t. text prompts, we render 9 sudoku-like novel views around each generated scene $\mathcal{E}_i, 1 \leq i \leq 4$, and compute the CLIP (Radford et al., 2021) scores of the prompt versus the rendered images. (2) We evaluate rendered novel view image quality with CLIP-IQA+ (Wang et al., 2023), Q-align IQA (Wu et al., 2024), and NIQE (Mittal et al., 2013). (4) We also measure the aesthetics of novel views by the Q-align IAA (Wu et al., 2024). We leave more details in the supplementary material.

**Implementation details.** In our implementation, we use Chain-of-Zoom (Kim et al., 2025) as our super-resolution model. We use Gen3C (Ren et al., 2025) as the camera-controlled video diffusion model in auxiliary view synthesis. We estimate image depth by MoGe (Wang et al., 2024a) and video depth by GeometryCrafter (Xu et al., 2025). We leave more details in the supplementary material. We will release the full code and software for reproducibility.

### 4.1 COMPARISON

**Qualitative showcase.** We show qualitative comparison in Figure 3 as well as Figures 7 and 8 in the supplementary material. We also strongly encourage the reader to see video results and to

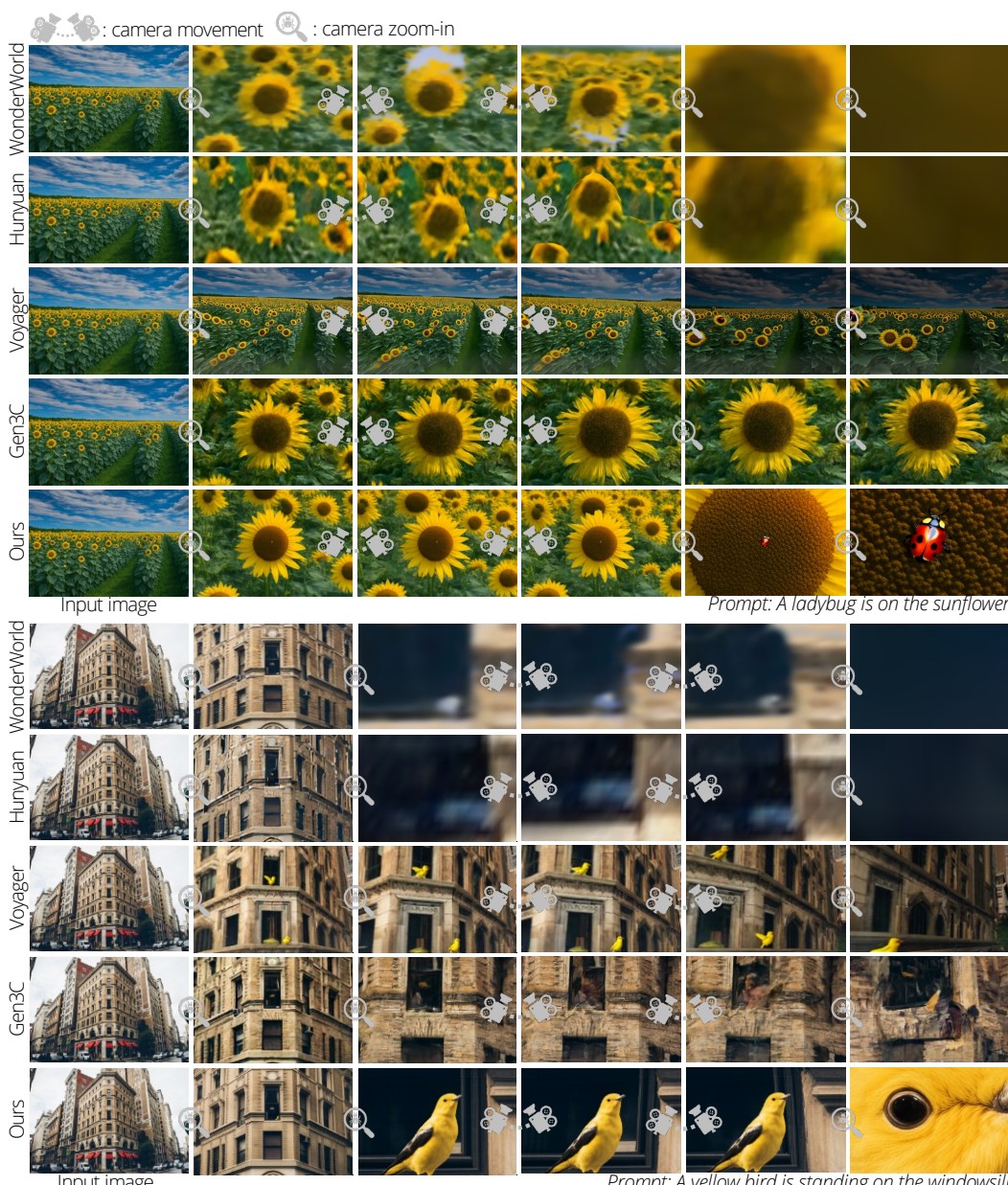

Figure 3: Visual comparison of our WonderZoom and baselines on multi-scale 3D world generation.

| Method | CS↑ | CIQA↑ | QIQA↑ | NIQE↓ | QIAA↑ | Time/s |
|---|---|---|---|---|---|---|
| WonderWorld (Yu et al., 2025) | 0.2691 | 0.5060 | 1.084 | 21.78 | 1.336 | **9.3** |
| HunyuanWorld (Team et al., 2025) | 0.2504 | 0.2820 | 1.054 | 15.23 | 1.306 | 704.2 |
| Gen3C (Ren et al., 2025) | 0.3010 | 0.5498 | 2.998 | 4.918 | 2.016 | 306.7 |
| Voyager (Huang et al., 2025) | 0.2603 | 0.5754 | 3.152 | 4.919 | 2.934 | 596.6 |
| WonderZoom (Ours) | **0.3427** | **0.7028** | **3.921** | **3.690** | **2.981** | 62.1 |

Table 1: Quantitative comparison. "CS" denotes CLIP score, "CIQA" denotes CLIP-IQA+, "QIQA" denotes Q-align IQA, "QIAA" denotes Q-align IAA. "Time" measures the time used in generating a new scale scene.

interactively view generated worlds on our website[1]. From the qualitative comparison, we find that the state-of-the-art 3D scene generation methods and the controllable video generation methods are not able to create multi-scale scenes. In particular, 3D methods always generate blurry zoom-in views as their 3D scene representations (i.e., Gaussian surfels in WonderWorld (Yu et al., 2025) and meshes

---

[1] https://wonderzoom.github.io/

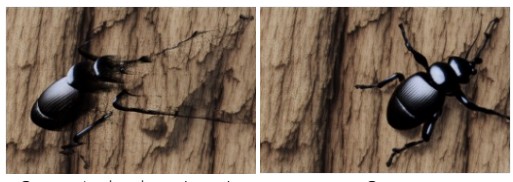

Ours w/o mod                     Ours

Figure 4: Ablation on the opacity modulation.

| Methods | Metrics | GPU Mem | FPS |
|---|---|---|---|
| Ours w/o mod. | | 7.96G | 1.4 |
| Ours | | **3.40G** | **97.2** |

Table 2: Comparison of computational cost for variants about scale-adaptive opacity modulation.

Ours w/o depth registration          Ours

Figure 5: Ablation study on our depth registration.

Ours w/o auxiliary view          Ours

Figure 6: Ablation study on auxiliary view synthesis.

in HunyuanWorld (Team et al., 2025)) do not support dynamic updating when new scale images are generated. Camera-controlled video models are able to zoom in, yet their control is imprecise compared to explicit 3D methods, and their generated views are not aligned with the prompts. In contrast, WonderZoom allows creating new scale structures that are closely aligned with the prompts, and generates high-quality novel views at any new scale.

**Quantitative comparison.** We show the quantitative metrics in Table 1. WonderZoom outperforms all baseline methods in terms of alignment, novel view quality, as well as aesthetics metrics. This further validates our observations through visual comparison.

## 4.2 ABLATION STUDY

We evaluate how the key technical components affect the multi-scale generation performances. We focus on the scale-aware opacity modulation, depth registration, and auxiliary view synthesis.

**Scale-aware opacity modulation.** We consider a variant "Ours w/o mod." which removes our scale-aware opacity modulation. We show a visual comparison in Figure 4 and a quantitative comparison on computational cost in Table 2. From the table, we can see that without our scale-aware opacity modulation, the computational burden makes it intractable for multi-scale real-time rendering. Furthermore, we observe from the visual result that it creates blurry renderings due to the lack of an appropriate mechanism for rendering multi-scale surfels. In contrast, ours maintains a high-quality rendering while requiring lower GPU memory and providing much faster rendering speed.

**Depth registration.** We consider a variant "Ours w/o depth registration" that removes the scale-consistent depth registration from WonderZoom. We show a visual comparison in Figure 5. As we can see in the comparison, removing our depth registration creates significant shape distortion on the new detail depth estimation, i.e., the newly generated beetle is distorted when observed from novel views. Our depth registration significantly alleviates this artifact.

**Auxiliary view synthesis.** We compare our model with "Ours w/o auxiliary view". As shown in Figure 6, our auxiliary view synthesis is critical in generating a complete 3D scene, while removing it leads to missing regions as revealed by the grey areas.

## 5 CONCLUSION

We presented WonderZoom which allows multi-scale 3D world generation from a single image. Through the scale-adaptive Gaussian surfels and a progressive detail synthesizer, we enable users to interactively zoom into any region and synthesize entirely new details while maintaining cross-scale consistency and real-time rendering. Our experiments demonstrate significant improvements over existing 3D-based and video-based methods in both visual quality and prompt alignment. WonderZoom opens new possibilities for interactive content creation and virtual world exploration across multiple orders of magnitude in scale.

## REPRODUCIBILITY STATEMENT

To ensure the reproducibility of our work, we have made significant efforts to provide comprehensive implementation details and will release all necessary resources. We will release the full source code and interactive demo software upon publication, enabling researchers to reproduce our multi-scale 3D world generation results and build upon our framework. The complete algorithmic procedure for our approach is detailed in Algorithm 1, which provides a step-by-step description of both the real-time rendering thread and the progressive detail synthesis pipeline. Our scale-adaptive Gaussian surfel representation is mathematically formulated in Section 3.1 and the explanation of seamless scale transition (Proposition 1). Optimization hyperparameters are specified in the main text, including the photometric loss function, the Adam optimizer settings, and the surfel initialization parameters. The specific external models used in our pipeline (VLM for semantic extraction, super-resolution model, controlled image editing model, monocular depth estimator, SAM for segmentation, and video diffusion model) are identified in the implementation details section. We will release test examples with corresponding input images, camera trajectories, and user prompts.

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

# A   APPENDIX

In the appendix, we provide an algorithm of WonderZoom in Alg. 1, additional visual comparison results in Figure 7 and Figure 8, additional implementation details, and LLM usage statement.

**Additional implementation details.** All images are processed at a resolution of $720 \times 1088$. We use GPT-4V as our VLM for semantic context extraction and editing prompt generation. The initial camera focal length is set to $f_x = f_y = 1024$, with progressive zoom-in operations increasing the focal length for finer scales, typically we multiply the current focal length by $8$ for a new scale. We use INR-Harmonization (Chen et al., 2023) after image editing for improved shading consistency.

**LLM usage statement.** In the preparation of this manuscript, we used large language models (LLMs) solely as a writing assistance tool for grammatical error correction and language refinement. Specifically, LLMs were employed only to improve the clarity and fluency of our existing text, correct grammatical mistakes, and enhance the readability of technical descriptions. All scientific contributions, including the core ideas, technical innovations, experimental design, analysis, and insights presented in this paper, were conceived and developed entirely by the authors without any LLM assistance. The LLMs did not generate any significant new text, contribute to the ideation process, or influence the scientific content or conclusions of our work. The conceptualization of scale-adaptive Gaussian surfels, the progressive detail synthesizer, the multi-scale generation framework, and all experimental analyses represent original work by the authors. We take full responsibility for the accuracy and integrity of all content presented in this paper.

---

**Algorithm 1** Multi-Scale 3D World Generation Control Loop

---

**Input:** Initial image $\mathbf{I}_0$, initial camera $\mathbf{C}_0 \in \mathbb{R}^{4\times4}$
**Output:** Multi-scale scene hierarchy $\{\mathcal{E}_0, \mathcal{E}_1, \ldots, \mathcal{E}_n\}$
**Runtime output:** Real-time rendered observation $\mathbf{O}_{\text{render}}$
**Runtime user control:** Camera viewpoint $\mathbf{C}_{\text{render}}$, zoom region $\mathbf{C}_{i+1}$, (optional) edit prompt $\mathcal{U}_{i+1}$

1: **Initialize:** $\mathcal{E}_0 \leftarrow \text{ReconstructScene}(\mathbf{I}_0, \mathbf{C}_0)$                    ▷ Initial 3D scene from input image
2: $\mathbf{C}_{\text{render}} \leftarrow \mathbf{C}_0$                    ▷ Initialize rendering camera
3: $i \leftarrow 0$                    ▷ Current scale index

4: **Thread 1: Real-time Scale-Adaptive Rendering**                    ▷ Continuous rendering loop
5: **while** true **do**
6:       $s^{\text{render}} \leftarrow d^{\text{render}} / \sqrt{f_x^{\text{render}} f_y^{\text{render}}}$                    ▷ Compute rendering scale
7:       $\mathbf{O}_{\text{render}} \leftarrow \text{RenderWithOpacityModulation}(\bigcup_{k=0}^{i} \mathcal{E}_k, \mathbf{C}_{\text{render}})$                    ▷ Sec. 3.1
8:       $\mathbf{C}_{\text{render}} \leftarrow \text{UserCameraControl}()$                    ▷ Interactive camera update
9: **end while**

10: **Thread 2: Progressive Detail Synthesis**    ▷ Triggered by user zooming into region of interest
       with prompt $\mathcal{U}_{i+1}$ at camera $\mathbf{C}_{i+1}$
11: *// Stage 1: New Scale Image Synthesis*
12: $\mathbf{O}_{i+1} \leftarrow \text{Render}(\mathcal{E}_i, \mathbf{C}_{i+1})$                    ▷ Coarse observation at zoomed view
13: $\mathcal{S} \leftarrow \text{VLM}(\text{Render}(\mathcal{E}_i, \mathbf{C}_i))$                    ▷ Extract semantic context
14: $\mathbf{I}'_{i+1} \leftarrow \text{SuperResolution}(\mathbf{O}_{i+1}, \mathcal{S})$                    ▷ Extreme super-resolution
15: **if** $\mathcal{U}_{i+1} \neq \emptyset$ **then**
16:       $\mathbf{I}_{i+1} \leftarrow \text{ControlledEdit}(\mathbf{I}'_{i+1}, \mathcal{U}_{i+1})$                    ▷ Insert user-specified content
17: **else**
18:       $\mathbf{I}_{i+1} \leftarrow \mathbf{I}'_{i+1}$
19: **end if**
20: *// Stage 2: Scale-Consistent Depth Registration*
21: $\mathbf{D}_{i+1}^{\text{target}} \leftarrow \text{RenderDepth}(\mathcal{E}_i, \mathbf{C}_{i+1})$                    ▷ Target depth from coarse scale
22: $\mathbf{D}_{i+1} \leftarrow \text{DepthRegistration}(\mathbf{I}_{i+1}, \mathbf{D}_{i+1}^{\text{target}})$                    ▷ Fine-tune depth estimator
23: *// Stage 3: Scale-Adaptive Surfel Generation*
24: $\mathcal{E}_{i+1}^{\text{partial}} \leftarrow \text{InitializeSurfels}(\mathbf{I}_{i+1}, \mathbf{D}_{i+1}, \mathbf{C}_{i+1})$
25:                    ▷ Create surfels with $s^{\text{native}} = d^{\text{native}} / \sqrt{f_x^{\text{native}} f_y^{\text{native}}}$
26: *// Stage 4: Auxiliary View Synthesis*
27: $\{\mathbf{C}_{i+1}^k\}_{k=1}^K \leftarrow \text{SampleNeighboringViews}(\mathbf{C}_{i+1})$
28: $\{\mathbf{I}_{i+1}^k, \mathbf{D}_{i+1}^k\} \leftarrow \text{AuxiliaryViewSynthesis}(\mathcal{E}_{i+1}^{\text{partial}}, \{\mathbf{C}_{i+1}^k\})$
29: *// Stage 5: Optimization*
30: $\mathcal{E}_{i+1} \leftarrow \text{OptimizeSurfels}(\mathcal{E}_{i+1}^{\text{partial}}, \{\mathbf{I}_{i+1}, \mathbf{I}_{i+1}^1, \ldots, \mathbf{I}_{i+1}^K\})$
31:                    ▷ Optimize $\{\mathbf{q}, \mathbf{s}, o\}$ with $\mathcal{L} = 0.8L_1 + 0.2L_{\text{D-SSIM}}$
32: $i \leftarrow i + 1$                    ▷ Increment scale index

---

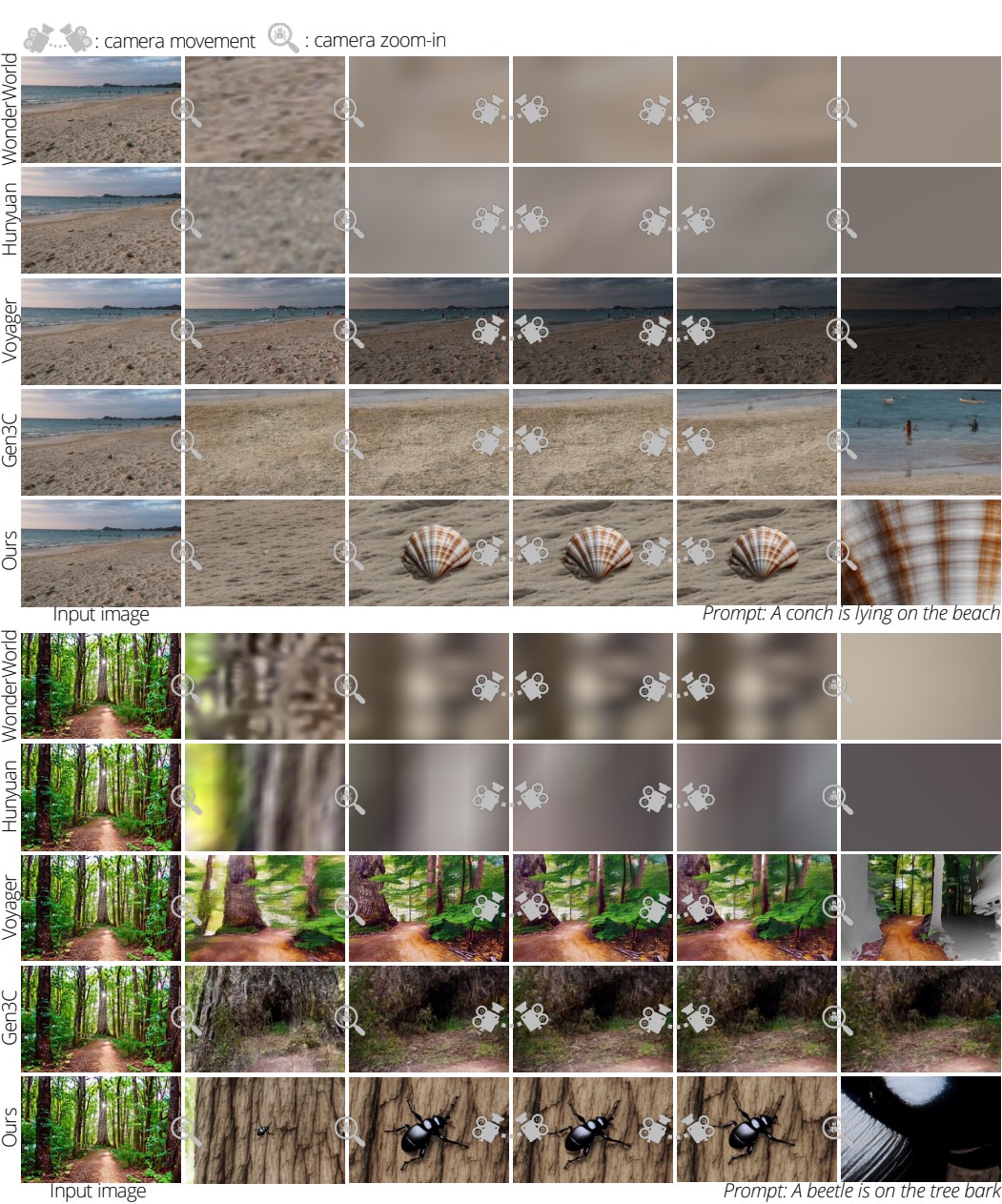

Figure 7: Visual comparison of multi-scale 3D world generation results.

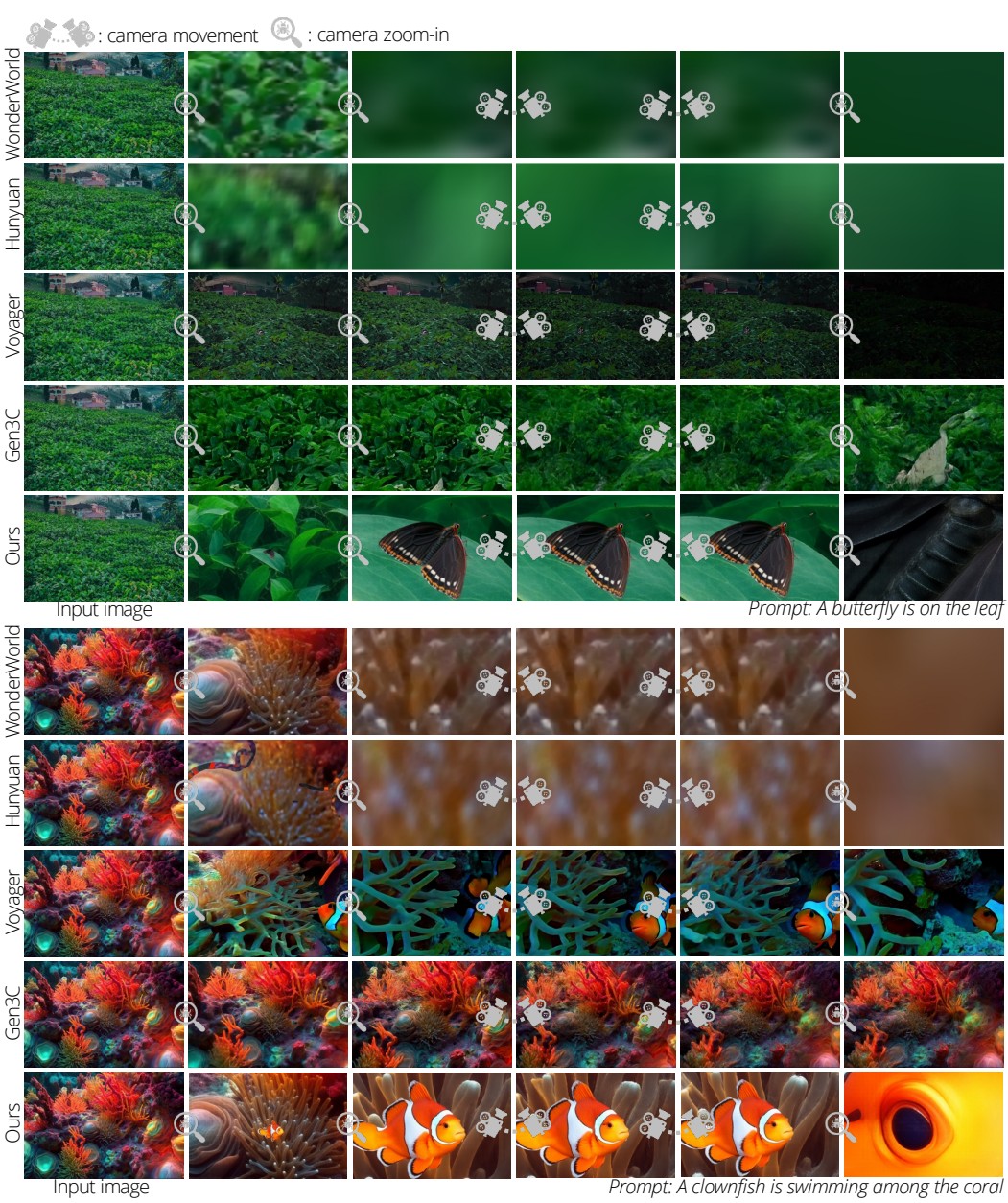

Figure 8: Visual comparison of multi-scale 3D world generation results.

