# OpenReview forum: "WonderZoom:  Multi-Scale 3D World Generation"
_ICLR.cc/2026/Conference — ICLR 2026 Conference Withdrawn Submission_

### Official Review · Reviewer_KTZf · 2025-10-26

**Soundness:** 3
**Presentation:** 3
**Contribution:** 3
**Rating:** 4
**Confidence:** 3

**Summary:**

WonderZoom is a framework for generating multi-scale 3D scenes from a single image. The method introduces scale-adaptive Gaussian surfels, which support dynamic updates and real-time rendering. It further proposes a progressive detail synthesizer consisting of three components: new-scale image generation, depth registration, and auxiliary view generation. Experimental results demonstrate that WonderZoom outperforms existing baselines.

**Strengths:**

1. Novel problem setting. The task of multi-scale 3D scene generation is both original and practical. Previous methods typically focus on single-scale generation, limiting their ability to support continuous exploration.

2. The proposed scale-adaptive Gaussian surfels and progressive detail synthesizer are well-motivated and empirically effective.

**Weaknesses:**

1. The experiments are conducted on only six scenes, which is insufficient to demonstrate robustness or generalization. It is unclear how representative these examples are, and whether they might be cherry-picked. A broader and more systematic evaluation would strengthen the paper.

2. A user study or at least a user-oriented evaluation would help substantiate claims about interactive quality.

3. The paper does not provide any failure cases or discussions of limitations.

**Questions:**

1. How robust is the overall pipeline, given that it relies on multiple base models? Under what conditions does it fail?

2. Can the next scale be generated via camera movements other than zoom-in (e.g., panning left/right or upward/downward)? All examples shown seem to rely solely on zoom-in motion.

---

### Official Review · Reviewer_SZuN · 2025-10-27

**Soundness:** 3
**Presentation:** 4
**Contribution:** 3
**Rating:** 6
**Confidence:** 4

**Summary:**

This paper presents WonderZoom, a novel framework designed for multi-scale 3D world generation from a single image. Unlike existing 3D scene generation systems—such as WonderWorld, HunyuanWorld, and SceneScape—which typically produce content at a fixed scale like landscapes, rooms, or objects, WonderZoom enables the creation of consistent 3D content across multiple spatial scales.

To address this challenge, the framework introduces two key technical contributions: Scale-Adaptive Gaussian Surfels, a hierarchical 3D representation that can be dynamically updated to incorporate fine-scale details without re-optimizing existing geometry, and the Progressive Detail Synthesizer, an autoregressive module that incrementally generates fine-scale 3D structures guided by user prompts and camera zoom-ins while preserving both geometric and semantic consistency.

With WonderZoom, users can interactively zoom into regions of a generated 3D scene to create new structures—including microscopic details—that were not present at broader scales. Experimental results demonstrate that WonderZoom achieves superior performance compared to leading 3D and video generation methods in terms of prompt alignment, rendering quality, and visual aesthetics.

**Strengths:**

1.  **Novel Problem Formulation**
    This work presents a significant paradigm shift by being the first to explicitly tackle **multi-scale 3D generation**. It moves beyond the established challenges of reconstruction or single-scale generation, pioneering a new direction for interactive and progressive 3D synthesis. This is a highly compelling and timely contribution.

2.  **Innovative Technical Representation**
    The introduction of **scale-adaptive Gaussian surfels** is an original and well-motivated representation. By ingeniously extending Gaussian surfels to be scale-aware with dynamic update mechanisms, it effectively bridges the domains of high-fidelity rendering and generative modeling.

3.  **Comprehensive Empirical Validation**
    The experimental section is thorough and demonstrates that WonderZoom consistently outperforms existing baseline methods in the challenging task of multi-scale 3D world generation. The results robustly validate the effectiveness of the proposed approach.

4.  **Exemplary Clarity and Structure**
    The paper is exceptionally well-organized and clearly written. The logical flow and structured presentation make the complex technical contributions easy to understand and follow.

5.  **Strong Potential Impact**
    WonderZoom holds considerable promise for advancing the frontier of 3D content creation. Its capability for generation at arbitrary scales has the potential to transform workflows in fields such as 3D design, virtual reality world-building, and scientific visualization.

**Weaknesses:**

1.  **Multi-Scale Consistency in Texture and Content**
    The rendered examples reveal challenges in maintaining consistent detail and texture across different scales. Spatial transitions often appear uneven when content from various scales is composited. For instance:
    *   In the **beach scene (Scale 1)**, the central region is rendered with significantly finer detail than the periphery, creating a noticeable and artificial boundary.
    *   In the **yellow bird example (Scale 1)**, the textures of the background roof eaves are visibly misaligned.
    Such inconsistencies ultimately undermine the realism and aesthetic coherence of the final output.

2.  **Limitations in Quantitative Evaluation**
    While the visual comparisons are comprehensive, the quantitative evaluation feels limited. For example, the analysis lacks user study results to statistically validate the perceptual quality of the outputs.

    Furthermore, the paper proposes that **Proposition 1 (Seamless Scale Transition)** facilitates smoother transitions, but this claim is not yet substantiated by experimental data. Providing metrics or a comparative analysis to demonstrate this improvement would strengthen the argument significantly.

**Questions:**

1.  **Error Accumulation in the Two-Stage Process**
    Your approach employs a two-stage method for generating finer-scale images: first applying super-resolution, followed by image editing. Could this sequential process lead to an accumulation of errors, ultimately causing inconsistencies in style and texture between the newly generated output and the existing scene?

2.  **Consistency in Overlapping Regions**
    During the iterative process of zooming into and out of different regions, if these areas overlap, what is the model's behavior? Will it strive to preserve the details generated in a previous pass, or will it overwrite them, potentially leading to temporal inconsistencies in the overlapping sections?

---

### Official Review · Reviewer_PcQR · 2025-10-30

**Soundness:** 3
**Presentation:** 4
**Contribution:** 3
**Rating:** 6
**Confidence:** 4

**Summary:**

The paper presents WonderZoom, a framework for generating multi-scale 3D worlds from a single image. It allows users to zoom into any region and synthesize new, coherent fine-scale content. The method introduces scale-adaptive Gaussian surfels for dynamic, real-time 3D representation and a progressive detail synthesizer that adds finer geometry and appearance guided by user input. Experiments demonstrate improved visual quality, consistency, and prompt alignment compared to existing 3D and video generation methods.

**Strengths:**

1. The paper tackles an important and underexplored problem of generating coherent 3D worlds across multiple spatial scales, demonstrating clear novelty compared to existing single-scale approaches.
2. The proposed scale-adaptive Gaussian surfels provide an efficient representation that enables dynamic updates and real-time rendering without re-optimization.
3. Quantitative experiments show consistent improvements over several strong baselines in prompt alignment and perceptual quality, supporting the method's effectiveness.

**Weaknesses:**

1. The experimental dataset is limited in diversity and scale, making it uncertain whether the method generalizes to more complex or realistic scenes.
2. The paper could be strengthened by including an analysis of potential failure cases that may arise during repeated zoom-in generations, such as semantic drift or geometric inconsistency across scales, to better understand the model's robustness.
3. While the paper includes ablation studies on several key components, the contribution of each stage within the progressive detail synthesizer (e.g., super-resolution, editing, and depth registration) is not analyzed separately. A more detailed breakdown could help clarify how these stages jointly affect cross-scale coherence and visual quality.

**Questions:**

Refer to Weaknesses.

---

### Official Review · Reviewer_qch5 · 2025-10-30

**Soundness:** 3
**Presentation:** 3
**Contribution:** 2
**Rating:** 4
**Confidence:** 4

**Summary:**

WonderZoom achieves multi-scale 3d world generation which let the user to interactively zoom into one region of the scene, and explore contents. It proposes a scale-adaptive Gaussian surfels representation, which will selectively choose Gaussian surfels for rendering based on the current scale. Besides, for generative super resolution purpose, they propose a progressive refiner that add new zoomed-in details. And they perform test-time optimization over depth prior to ensure depth consistency across scales. Finally, VLM is used to support multi-view synthesis after getting into one zoom level. For experiments, the authors carefully compare their framework with several video-based generator (single-scaled), and showcases their quality is better. And they ablate their three design choices in terms of depth registration, scale-based selective Gaussian Surfel rendering, and multi-view auxiliary view with video prior.

**Strengths:**

1. It is the first work that can generates extreme-scale 3d exploration.

2. Generic and modular framework to support multi-scale scene generation, which is not restricted to any prior to use.

**Weaknesses:**

1. Lack of clear practical use cases.
While the proposed techniques are technically sound, their real-world utility remains unclear. The idea of generative, multi-modal content at extreme zooming scales (e.g., zooming into a flower to reveal a bug) appears visually interesting but lacks an evident practical motivation or application scenario.

2. Inconsistency across scales.
The generated content exhibits noticeable discontinuities between scales. As shown in the qualitative results, fine-scale details are not preserved when zooming out. The use of scale-dependent Gaussian surfels, though computationally efficient, undermines 3D structural consistency across scales.

3. Limited and unnatural interaction design.
The system does not support seamless, free-form interaction within a unified 3D scene. Users must explicitly select a scale level, after which the model hallucinates new content at that scale. This results in fragmented and unrealistic interactions—far from how people naturally perceive or explore multi-scale scenes.

4. Lack of architectural coherence.
The framework integrates multiple independent components without a strong unifying principle. This design introduces redundancy and heavy reliance on various priors. For instance, auxiliary view synthesis is treated as a separate generation process rather than being jointly optimized with depth prediction across scales.

5. Weak and incomplete baseline comparisons.
The paper compares against only trivial or loosely related baselines. Even though there are few existing 3D multi-scale generation methods, several strong 2D multi-scale models (e.g., EarthGen) could serve as relevant baselines. Incorporating simple depth priors into such 2D frameworks would likely yield more competitive multi-scale results.

**Questions:**

1. What would happen if we use all Gaussian surfels for rendering? Will there be artifacts due to content inconsistency across scales?

2. Why is it one interesting task for the community to explore? what is the implication of such generative super resolution task?

---

### Note · Authors · 2025-11-12

**Comment:**

Thanks for all the constructive comments.

**Withdrawal Confirmation:**

I have read and agree with the venue's withdrawal policy on behalf of myself and my co-authors.